# Patient satisfaction in regional referral hospitals of Bhutan: Insights from a cross-sectional study

**Kuenzang Dorji**[1]*, **Kinga Jamphel**[2], **Jigme Kelzang**[1], **Ugyen Pelmo**[3‡], **Hem Kumar Nepal**[4‡], **Sonam Zangpo**[1‡], **Sonam Wangdi**[5‡], **Karma Galey**[1]

**1** Health Service Quality Assurance Division, Department of Health Service, Ministry of Health, Royal Government of Bhutan, Thimphu, Bhutan, **2** Department of Health Service, Ministry of Health, Royal Government of Bhutan, Thimphu, Bhutan, **3** Quality Assurance Unit, Gelephu Central Regional Referral Hospital, National Medical Services, Ministry of Health, Royal Government of Bhutan, Gelephu, Bhutan, **4** Quality Assurance Unit, Mongar Eastern Regional Referral Hospital, National Medical Services, Ministry of Health, Royal Government of Bhutan, Mongar, Bhutan, **5** Policy Planning Division, Secretariat, Ministry of Health, Royal Government of Bhutan, Thimphu, Bhutan

ォ These authors contributed equally to this work.
‡ UP, HN, SZ and SW also contributed equally to this work.
* kuenzangdbiomed@gmail.com (KD)

## Abstract

### Background

Patient satisfaction is crucial for evaluating healthcare quality and guiding continuous quality improvement. Globally, patient satisfaction has been extensively studied; however, there is limited research on this topic in Bhutan, where the healthcare system is in the early stages of developing a quality-oriented culture. To address this gap, we aimed to evaluate patient satisfaction levels among different socio-demographic and clinical groups and identify the predictors of patient satisfaction in Bhutan.

### Methods

We conducted a retrospective analysis of patient satisfaction survey responses archived in the quality assurance unit of two tertiary healthcare centres in Bhutan: Mongar Eastern Regional Referral Hospital and Gelephu Central Regional Referral Hospital. The routine surveys, administered throughout April 2024, utilised an adapted version of the Patient Satisfaction Questionnaire-18. The data were analysed using descriptive and inferential statistics.

### Results

Our study revealed significant variations in patient satisfaction across socio-demographic and clinical groups. Ethnicity *(P-value = 0.017)*, occupation *(P-value = 0.014)*, and education level *(P-value = 0.021)* emerged as significant predictors of satisfaction. Sharchop and other ethnic groups *(P-value = <0.001)*; farmers, religious personnel, and other occupational groups *(P-value=<0.001)*;

**Data availability statement:** Data cannot be shared publicly because it is owned by the Ministry of Health, Royal Government of Bhutan. Data are available from the Research Ethics Board of Health, Ministry of Health, Royal Government of Bhutan (contact via rebhscretary@gmail.com) for researchers who

meet the criteria for access to confidential data. The data underlying the results presented in the study are available from the Research Ethics Board of Health (https://www.moh.gov.bt/about/program-profiles/357-2/).

**Funding:** The author(s) received no specific funding for this work.

**Competing interests:** The authors have declared that no competing interests exist.

and illiterate *(P-value=<0.001)* individuals exhibited significantly higher satisfaction levels. While patient type *(P-value=0.472)*, age *(P-value=0.553)*, and marital status *(P-value=0.448)* influenced satisfaction levels, they did not emerge as significant predictors when considering other variables. Overall, patient satisfaction in Bhutan is 4.06 on a 5-point Likert scale. Satisfaction is highest in the financial domain, while accessibility and convenience received the lowest scores.

## Conclusions

Overall, with a score of 4.06 on a 5-point Likert scale, patient satisfaction in Bhutan is high. However, our findings highlight the need to address socio-demographic disparities in patient satisfaction. As the Bhutanese socio-demographic landscape evolves, satisfaction levels may decline. To enhance overall satisfaction, healthcare policymakers should focus on improving accessibility and convenience. Strategies such as establishing dynamic limits on free services, exploring private sector engagement in advanced healthcare service, and strengthening the healthcare workforce are essential for sustainable and quality healthcare service delivery.

## Introduction

Globally, the healthcare system is evolving to prioritise patient-centred care as a fundamental aspect of healthcare delivery and therefore meeting patient's needs is imperative. This shift has emphasized the necessity of placing a strong focus on fulfilling the patient's preferences, needs, and values to provide quality healthcare [1–3]. Since the early 1980s, there have been ongoing efforts to comprehend and measure patient satisfaction, and gradually, it has been acknowledged as a crucial component in delivering quality healthcare [4]. However, there is still no consensus on a universally accepted definition of patient satisfaction [5,6]. Patient satisfaction is influenced by several factors, such as technique, functionality, infrastructure, interaction, environment, and services, making it multidimensional and subjective [3,6,7]. Moreover, factors inherent to patients, such as age, education level, and health status, which healthcare professionals and managers cannot control, exert an influence on patient satisfaction, adding further complexity to the matter [3]. Despite the ambiguity, in many countries with advanced healthcare systems, patient satisfaction has been used as a standard measure of healthcare quality [8–10]. Assessing the quality of healthcare services from the patient's viewpoint is essential because they are the ultimate beneficiaries of the healthcare services [3]. Patient feedback can assist in recognizing unfulfilled patient needs, providing healthcare managers and professionals with valuable guidance for Continuous Quality Improvement (CQI) [7,10].

Bhutan, nestled between India and China, established its modern healthcare system in the 1960s. The present healthcare system is state-funded and structured into three tiers, offering free healthcare services to all citizens. It consists of national or regional referral hospitals at the tertiary level, district hospitals at the secondary level, and primary healthcare centres and outreach clinics at the primary level [11].

Over the years, Bhutan's healthcare system has made remarkable progress; however, it continues to face substantial challenges in delivering quality and safe services. These include overburdened staff, shortages of specialised professionals, limited financial resources restricting access to advanced equipment and medicines, and inadequate infrastructure. Additionally, geographical barriers such as rugged terrain and poor road connectivity further hinder access, especially in remote areas [12–15].

Since 2002, the Ministry of Health (MoH, Royal Government of Bhutan) has strengthened its emphasis on delivering quality and safe healthcare services. The MoH, in collaboration with the Bhutan Standard Bureau (Ministry of Industry, Commerce, and Employment; Royal Government of Bhutan), developed and published the Bhutan Healthcare Standard for Quality Assurance (BHSQA), a nation's first healthcare standard in 2018. The BHSQA contains 116 standards, 639 objective elements, and 67 key performance indicators covering both clinical and managerial structures, processes, and outcomes. To raise the quality and safety of healthcare services to the desired level, the Health Service Quality Assurance Division of the MoH gradually implemented the BHSQA across all healthcare centres in the country [16,17].

At present, the Bhutanese healthcare system is in its early stages of developing a culture focused on quality and safety, necessitating constant vigilance and evaluation of its advancement. As a result, each healthcare centre across the nation is required to evaluate healthcare quality and safety using BHSQA key performance indicators, which include assessing patient satisfaction. To gauge patient satisfaction, Bhutanese healthcare centres utilise the Patient Satisfaction Questionnaire 18 (PSQ-18), a well-validated tool noted for its brevity and efficacy across various contexts [10,18,19]. However, due to various challenges, published reports on patient satisfaction in Bhutan are scarce. This deficiency hampers the development and implementation of targeted interventions necessary for establishing an effective, efficient, and responsive healthcare system. To address this gap, this study aims to evaluate patient satisfaction levels among different socio-demographic and clinical groups and identify the predictors of patient satisfaction in Bhutan.

## Materials and methods

### Study design and setting

This retrospective cross-sectional study analysed archived patient satisfaction survey data collected from Mongar Eastern Regional Referral Hospital (MERRH) and Gelephu Central Regional Referral Hospital (GCRRH). The hospitals had previously conducted routine paper-based surveys throughout April 2024 as part of their CQI initiatives. During this period, voluntary feedback was sought from every fifth inpatient and outpatient, aged 18 years and older. A systematic selection of every fifth patient was employed to minimize selection bias and meet the hospital's predefined sample size requirements for internal evaluation. For illiterate participants, responses were recorded by their friends or quality assurance officials to ensure inclusivity. The completed surveys were briefly analysed for CQI and archived in the respective quality assurance unit.

### Study population and survey instrument

The study population consisted of patients aged 18 years and older who had voluntarily completed surveys. The original survey employed an adapted version of the PSQ-18, comprising two sections: patient details and satisfaction indicators. The patient details section collected eight socio-demographic and clinical variables across 33 subgroups. The satisfaction indicators section assessed seven domains of patient satisfaction as outlined in the standard PSQ-18. To ensure broader applicability across healthcare settings, the term "doctor" or "doctor's office" was replaced with "healthcare professional" or "healthcare centre" respectively. Responses were recorded using a 5-point Likert scale: 1 (strongly agree), 2 (agree), 3 (uncertain), 4 (disagree), and 5 (strongly disagree).

The PSQ-18 consists of both negative and positively constructed questions, and therefore, to address inconsistencies in interpretation, the original Likert scale responses were re-scaled again on a 5-point Likert scale to ensure uniformity. This re-scaling process adhered to the standard PSQ-18 conversion, where higher scores indicate greater satisfaction with healthcare services.

## Data management and analysis

All archived data were retrieved, reviewed for completeness, and incomplete surveys were excluded. To assess differences in patient satisfaction levels among predefined groups in the survey, a one-way analysis of variance (ANOVA) was conducted. Subsequently, Dunnett's test was applied to identify statistically distinct groups within these predefined categories. Depending on which provided the most interpretable results, the group with the lowest or the highest mean score was used as the comparison reference. Effect size estimates (Cohen's d or f) were calculated to quantify the magnitude of group differences. A post-hoc evaluation of statistical power was also performed to assess the adequacy of the sample size in detecting meaningful differences.

The overall patient satisfaction score of individual patients, calculated as the average of all seven domain scores, was dichotomised into two categories: satisfied (scores above three) and dissatisfied (scores of three and below). Subsequently, binary logistic regression was performed to determine predictors of patient satisfaction among the newly established groups. The significance level was set at less than 0.05 for all statistical analyses. Descriptive statistics, including mean and standard deviation, were computed for each domain of patient satisfaction within the newly established statistically distinct groups. Data analysis was carried out using Minitab statistical software (version 17.1) and G*Power (version 3.1.9.7).

## Ethical considerations

The study was approved by the Research Ethics Board for Health, Ministry of Health, Royal Government of Bhutan. The archived data was accessed on 15th July 2024, and the authors did not have access to any information that could identify individual participants during or after data collection.

## Results

A total of 471 survey responses were collected from MERRH and 454 from GCRRH. After a thorough review, six survey responses from MERRH and four from GCRRH were excluded due to incomplete data, resulting in 915 survey responses for the final analysis. The socio-demographic composition of the respondents is presented in Table 1.

Among the survey respondents, 65.79% were outpatients and 34.21% were inpatients, with the majority (70.82%) aged between 18–44 years. Sex-wise, 56.72% of respondents were female and 43.28% were male, with 68.42% using healthcare services more than once. Regarding ethnicity, the majority of the respondents (57.05%) were Sharchop, while married individuals constituted the largest marital group (74.86%). The largest occupational group consisted of farmers (29.29%). Educationally, 45.58% had secondary education or lower, 26.01% were illiterate, and the remainder held various qualifications. The one-way ANOVA analysis showed significant differences in overall patient satisfaction levels across different socio-demographic and clinical groups (Table 2).

Patient satisfaction levels exhibited significant differences between inpatients and outpatients *(P-value=<0.001)*, with inpatients reporting higher satisfaction levels. Similarly, age was also identified as a significant factor influencing patient satisfaction level *(P-value=<0.001)*. Subsequent Dunnett multiple comparisons, using the 18–44 years age group as the control, showed that patient satisfaction levels for individuals aged 65 years and older differed significantly from the control group's mean.

Ethnicity also significantly influenced patient satisfaction levels *(P-value=<0.001)*. The comparisons, using the Ngalop group as the control, demonstrated significant differences in patient satisfaction levels for Sharchop and other ethnic groups compared to the control group's mean. Furthermore, marital status emerged as another significant factor influencing patient satisfaction level *(P-value=<0.001)*. Comparisons with the married group as a control indicated that only the unmarried group significantly differed from the control group's mean.

We also observed significant *(P-value=<0.001)* differences in satisfaction levels among occupational groups. Comparisons with civil servants as a control showed significant differences from the control level mean for farmers,

**Table 1. Socio-demographic and clinical profile of the respondents.**

| Variables | Groups | GCRRH (n = 450) | MERRH (n = 465) | Total (n = 915) |
|---|---|---|---|---|
| Patient type | Inpatient | 150 (33.33%) | 163 (35.05%) | 313 (34.21%) |
| | Outpatient | 300 (66.67%) | 302 (64.95%) | 602 (65.79%) |
| Age | 18-44 years | 315 (70.00%) | 333 (71.61%) | 648 (70.82%) |
| | 45-64 years | 104 (23.11%) | 85 (18.28%) | 189 (20.66%) |
| | 65-74 years | 22 (4.89%) | 21 (4.52%) | 43 (4.70%) |
| | ≥75 years | 9 (2.00%) | 26 (5.59%) | 35 (3.83%) |
| Sex | Male | 179 (39.78%) | 217 (46.67%) | 396 (43.28%) |
| | Female | 271 (60.22%) | 248 (53.33%) | 519 (56.72%) |
| Service utilisation | Once | 156 (34.67%) | 133 (28.60%) | 289 (31.59%) |
| | More than once | 294 (65.33%) | 332 (71.4%) | 626 (68.42%) |
| Ethnic group | Ngalop | 43 (9.56%) | 27 (5.81%) | 70 (7.65%) |
| | Lhotshampa | 195 (43.33%) | 14 (3.01%) | 209 (22.84%) |
| | Sharchop | 146 (32.44%) | 376 (80.86%) | 522 (57.05%) |
| | Others | 66 (14.67%) | 48 (10.32%) | 114 (12.46%) |
| Marital status | Unmarried | 70 (15.56%) | 109 (23.44%) | 179 (19.56%) |
| | Married | 357 (79.33%) | 328 (70.54%) | 685 (74.86%) |
| | Divorced | 15 (3.33%) | 18 (3.87%) | 33 (3.61%) |
| | Widowed | 8 (1.78%) | 10 (2.15%) | 18 (1.97%) |
| Occupation | Religious personnel | 4 (0.89%) | 10 (2.25%) | 14 (1.53%) |
| | Corporate | 15 (3.33%) | 4 (0.86%) | 19 (2.08%) |
| | Student | 21 (4.67%) | 56 (12.04%) | 77 (8.42%) |
| | Business | 98 (21.78%) | 48 (10.32%) | 146 (15.96%) |
| | Civil servant | 76 (16.89%) | 84 (18.06%) | 160 (17.49%) |
| | Farmer | 80 (17.78%) | 188 (40.43%) | 268 (29.29%) |
| | Others | 156 (34.67%) | 75 (16.13%) | 231 (25.24%) |
| Education level | Illiterate | 91 (20.22%) | 147 (31.61%) | 238 (26.01%) |
| | Non-formal education | 23 (5.11%) | 21 (4.52%) | 44 (4.81%) |
| | Secondary education or lower | 228 (50.67%) | 189 (40.65%) | 417 (45.58%) |
| | Certificate | 14 (3.11%) | 15 (3.23%) | 29 (3.17%) |
| | Diploma | 23 (5.11%) | 33 (7.09%) | 56 (6.12%) |
| | Bachelor | 54 (12.00%) | 34 (7.31%) | 88 (9.62%) |
| | Master | 3 (0.60%) | 8 (1.72%) | 11 (1.20%) |
| | Other qualification | 14 (3.11%) | 18 (3.87%) | 32 (3.50%) |

religious personnel, and others. Education level also played a significant role in influencing patient satisfaction level (P-value=<0.001). Using the master's degree group as a control for comparisons, only the illiterate group showed a significant difference from the control level mean. Additionally, regarding center-wise comparisons, patients at MERRH reported significantly higher satisfaction levels than those at GCRRH (P-value =<0.001). The binary logistic regression analysis identified several significant predictors of patient satisfaction (Table 3).

Among the predictor variables, sex (P-value=0.034), ethnicity (P-value=0.017), occupation (P-value=0.014), and education level (P-value=0.021) were found to be significant predictors of satisfaction among Bhutanese patients. Female patients have higher odds of satisfaction (odds ratio=1.67, 95% CI: 1.04–2.69). Similarly, patients from Sharchop and other ethnic backgrounds have higher odds of satisfaction (odds ratio=1.80, 95% CI: 1.12–2.91). Furthermore, patients involved in farming, religious services, or other occupations showed higher odds of satisfaction (odds ratio=1.95, 95%

**Table 2. One-way ANOVA and t-test analysis of patient satisfaction (average score) across socio-demographic and clinical variables.**

| Variables | Groups | Mean score (SD) | P-value | Effect size | Power (1−β) |
|---|---|---|---|---|---|
| Patient type‡ | Inpatient (n=313) | 4.14 (0.73) [a] | <0.001 | d=0.28 | 0.98 |
| | Outpatient (n=602) | 3.95 (0.62) | | | |
| Age† | 18-44 years (n=647) | 3.96 (0.68) [a] | <0.001 | f=0.18 | 1.00 |
| | 45-64 years (n=189) | 4.04 (0.65) [a] | | | |
| | 65-74 years (n=43) | 4.35 (0.51) | | | |
| | ≥75 years (n=36) | 4.44 (0.50) | | | |
| Sex‡ | Male (n=396) | 3.98 (0.71) [a] | 0.225 | d=0.08 | 0.23 |
| | Female (n=519) | 4.04 (0.63) [a] | | | |
| Service utilisation‡ | Once (n=289) | 4.00 (0.70) [a] | 0.674 | d=0.03 | 0.07 |
| | More than once (n=626) | 4.02 (0.65) [a] | | | |
| Ethnic group† | Ngalop (n=70) | 3.76 (0.63) [a] | <0.001 | f=0.13 | 0.94 |
| | Lhotshampa (n=209) | 3.94 (0.66) [a] | | | |
| | Sharchop (n=522) | 4.06 (0.67) | | | |
| | Others (n=114) | 4.06 (0.66) | | | |
| Marital status† | Unmarried (n=179) | 3.85 (0.67) | <0.001 | f=0.15 | 0.97 |
| | Married (n=685) | 4.07 (0.66) [a] | | | |
| | Divorced (n=33) | 3.80 (0.64) [a] | | | |
| | Widowed (n=18) | 4.10 (0.78) [a] | | | |
| Occupation† | Religious personnel (n=14) | 4.44 (0.45) | <0.001 | f=0.31 | 1.00 |
| | Corporate (n=19) | 3.66 (0.49) [a] | | | |
| | Student (n=77) | 3.89 (0.72) [a] | | | |
| | Business (n=146) | 3.82 (0.68) [a] | | | |
| | Civil servant (n=160) | 3.83 (0.71) [a] | | | |
| | Farmer (n=268) | 4.27 (0.55) | | | |
| | Others (n=231) | 4.03 (0.64) | | | |
| Education level † | Illiterate (n=238) | 4.29 (0.53) | <0.001 | f=0.29 | 1.00 |
| | Non-formal education (n=44) | 4.10 (0.72) [a] | | | |
| | Secondary education or lower (n=417) | 3.96 (0.65) [a] | | | |
| | Certificate (n=29) | 3.80 (0.85) [a] | | | |
| | Diploma (n=56) | 3.87 (0.68) [a] | | | |
| | Bachelor (n=88) | 3.70 (0.75) [a] | | | |
| | Master (n=11) | 3.77 (0.83) [a] | | | |
| | Other qualifications (n=32) | 4.01 (0.62) [a] | | | |
| Centre‡ | MERRH (n=465) | 4.17 (0.62) [a] | <0.001 | d=0.49 | 1.00 |
| | GCRRH (n=450) | 3.85 (0.67) | | | |

*Footnote* 'a' denote grouping information using the Dunnett method with 95% confidence. Means not labelled with the letter 'a' are significantly different from the control group's mean; ‡ indicates analysis performed using a 2-sample t-test; † refers to analysis performed using one-way ANOVA; 'd' represents Cohen's d (effect size for t-test), and 'f' denotes Cohen's f (effect size for ANOVA).

CI: 1.13–3.37), along with illiterate patients *(odds ratio = 2.65, 95% CI: 1.09–6.42)*. Table 4 displays the domain-specific patient satisfaction levels across different variables and groups.

The overall patient satisfaction, computed as an average of all seven domains, is 4.06 on a 5-point Likert scale. The Financial domain achieved the highest score with an average of 4.36, whereas the accessibility and convenience domain received the lowest score with an average of 3.75.

 

**Table 3. Analysis of patient satisfaction: Predictor effects using binary logistic regression.**

| Predictor variables | Adjusted Odds Ratio | 95% CI for Odds Ratio | P-value |
|---|---|---|---|
| **Patient type** | | | |
| Inpatient | 0.82 | 0.47-1.41 | 0.472 |
| Reference: Outpatient | | | |
| **Age** | | | |
| ≥65 years | 0.70 | 0.23- 2.18 | 0.553 |
| Reference: 18–64 years | | | |
| **Sex** | | | |
| Female | 1.67 | 1.04-2.69 | 0.034 |
| Reference: Male | | | |
| **Service utilisation** | | | |
| More than once | 1.31 | 0.79- 2.17 | 0.301 |
| Reference: Once | | | |
| **Ethnic group** | | | |
| Sharchop & Others | 1.80 | 1.12-2.91 | 0.017 |
| Reference: Ngalop & Lhotshampa | | | |
| **Marital status** | | | |
| Ever married | 1.24 | 0.71-2.17 | 0.448 |
| Reference: Unmarried | | | |
| **Occupation** | | | |
| Farmer, religious personnel & others | 1.95 | 1.13-3.37 | 0.014 |
| Reference: Civil servant, business, student and corporate | | | |
| **Education level** | | | |
| Illiterate | 2.65 | 1.09- 6.42 | 0.021 |
| Reference: Literate | | | |

## Discussion

Patient satisfaction, despite its multifaceted, dynamic, and subjective nature, is widely regarded as a significant indicator of healthcare quality. This recognition is grounded in the acknowledgement that patients are the primary beneficiaries of healthcare interventions, thus rendering their satisfaction a fundamental metric in evaluating the excellence of healthcare services [20–22]. Aligned with this perspective, Bhutan vigorously implements the BHSQA and routinely evaluates patient satisfaction to ensure optimal healthcare quality. The patient satisfaction survey data from MERRH and GCRRH includes patients from major Bhutanese socio-demographic groups. While the sample may not perfectly reflect the ethnic proportions of the broader population, it offers valuable insights into patient satisfaction across different groups, which can inform national healthcare policy and decision-making.

Our study showed that patients belonging to Sharchop and other ethnic groups; farmers, religious personnel, and other occupational groups, as well as illiterate groups, significantly exhibit higher levels of satisfaction and serve as significant predictors of satisfaction. However, the confidence interval of the regression model is wider, particularly for the education level group. This may be attributable to inherent variability or the effect of sample size, which could lead to less precise estimates.

Globally, patient satisfaction varies among different ethnic, occupational, and educational groups [20,21,23–36], highlighting the need for context-specific strategies to address disparities and improve overall patient satisfaction. Among Bhutanese ethnic groups, Sharchop individuals are generally considered sensitive, while Ngalops are perceived as assertive [37]. These differences in personality traits might have contributed to the difference in satisfaction levels observed between Sharchop and Ngalop individuals.

**Table 4. Descriptive statistics (mean, SD, minimum, and maximum) of patient satisfaction levels across seven domains within newly established, statistically distinct socio-demographic and clinical groups.**

| Variables | Domain 1 | Domain 2 | Domain 3 | Domain 4 | Domain 5 | Domain 6 | Domain 7 |
|---|---|---|---|---|---|---|---|
| **Patient type** | | | | | | | |
| Outpatient | 4.10±0.85 (1.0-5.0) | 3.93±0.79 (1.3-5.0) | 4.05±0.89 (1.0-5.0) | 4.09±0.87 (1.0-5.0) | 4.35±0.83 (1.0-5.0) | 3.93±0.94 (1.0-5.0) | 3.60±0.82 (1.0-5.0) |
| Inpatient | 4.15±0.91 (1.0-5.0) | 4.06±0.86 (1.0-5.0) | 4.24±0.96 (1.0-5.0) | 4.26±0.92 (1.0-5.0) | 4.38±0.91 (1.0-5.0) | 4.02±1.00 (1.0-5.0) | 4.04±0.84 (1.0-5.0) |
| **Age** | | | | | | | |
| 18-64 years | 4.10±0.88 (1.0-5.0) | 3.94±0.81 (1.0-5.0) | 4.08±0.93 (1.0-5.0) | 4.13±0.90 (1.0-5.0) | 4.33±0.87 (1.0-5.0) | 3.93±0.96 (1.0-5.0) | 3.70±0.85 (1.0-5.0) |
| ≥65 years | 4.31±0.80 (2.0-5.0) | 4.33±0.81 (1.8-5.0) | 4.56±0.63 (3.0-5.0) | 4.40±0.81 (2.0-5.0) | 4.67±0.60 (3.0-5.0) | 4.37±0.86 (1.5-5.0) | 4.28±0.77 (2.0-5.0) |
| **Sex** | | | | | | | |
| Male | 4.06±0.91 (1.0-5.0) | 3.91±0.82 (1.3-5.0) | 4.09±0.97 (1.0-5.0) | 4.14±0.91 (1.0-5.0) | 4.30±0.90 (1.0-5.0) | 3.91±1.00 (1.0-5.0) | 3.76±0.89 (1.0-5.0) |
| Female | 4.15±0.84 (1.5-5.0) | 4.02±0.81 (1.0-5.0) | 4.13±0.88 (1.0-5.0) | 4.16±0.88 (1.0-5.0) | 4.40±0.83 (1.0-5.0) | 4.01±0.93 (1.0-5.0) | 3.74±0.83 (1.0-5.0) |
| **Service utilisation** | | | | | | | |
| Once | 4.08±0.89 (1.3-5.0) | 3.97±0.82 (1.0-5.0) | 4.10±0.96 (1.0-5.0) | 4.14±0.91 (1.0-5.0) | 4.26±0.92 (1.0-5.0) | 3.89±0.97 (1.0-5.0) | 3.80±0.89 (1.0-5.0) |
| More than once | 4.13±0.86 (1.5-5.0) | 3.97±0.81 (1.0-5.0) | 4.12±0.90 (1.0-5.0) | 4.16±0.88 (1.0-5.0) | 4.40±0.82 (1.0-5.0) | 4.00±0.96 (1.0-5.0) | 3.72±0.84 (1.0-5.0) |
| **Ethnicity** | | | | | | | |
| Ngalop & Lhotshampa | 3.95±0.87 (1.0-5.0) | 3.80±0.78 (1.3-5.0) | 4.00±0.90 (1.0-5.0) | 3.96±0.90 (1.0-5.0) | 4.30±0.86 (1.0-5.0) | 3.80±0.98 (1.0-5.0) | 3.74±0.82 (1.0-5.0) |
| Sharchop & others | 4.19±0.87 (1.0-5.0) | 4.05±0.82 (1.0-5.0) | 4.16±0.92 (1.0-5.0) | 4.23±0.87 (1.0-5.0) | 4.38±0.86 (1.0-5.0) | 4.04±0.95 (1.0-5.0) | 3.75±0.87 (1.0-5.0) |
| **Marital status** | | | | | | | |
| Unmarried | 4.00±0.93 (1.5-5.0) | 3.77±0.84 (1.8-5.0) | 3.99±0.98 (1.0-5.0) | 4.03±0.89 (1.5-5.0) | 4.22±0.91 (1.0-5.0) | 3.85±0.96 (1.5-5.0) | 3.51±0.84 (1.0-5.0) |
| Ever married | 4.14±0.86 (1.0-5.0) | 4.02±0.80 (1.0-5.0) | 4.14±0.90 (1.0-5.0) | 4.18±0.89 (1.0-5.0) | 4.39±0.84 (1.0-5.0) | 3.99±0.96 (1.0-5.0) | 3.80±0.85 (1.0-5.0) |
| **Occupation** | | | | | | | |
| Farmer, religious personnel & others | 4.29±0.80 (1.0-5.0) | 4.16±0.76 (1.0-5.0) | 4.24±0.86 (1.0-5.0) | 4.27±0.84 (1.0-5.0) | 4.47±0.80 (1.0-5.0) | 4.16±0.91 (1.0-5.0) | 3.86±0.84 (1.0-5.0) |
| Civil servant, business, student & corporate | 3.90±0.92 (1.0-5.0) | 3.73±0.82 (1.0-5.0) | 3.95±0.96 (1.0-5.0) | 4.00±0.94 (1.0-5.0) | 4.22±0.91 (1.0-5.0) | 3.73±0.98 (1.0-5.0) | 3.60±0.85 (1.0-5.0) |
| **Education level** | | | | | | | |
| Illiterate | 4.36±0.74 (2.0-5.0) | 4.33±0.67 (2.3-5.0) | 4.40±0.72 (2.0-5.0) | 4.37±0.79 (1.0-5.0) | 4.55±0.78 (1.0-5.0) | 4.30±0.82 (2.0-5.0) | 4.00±0.77 (2.0-5.0) |
| Literate | 4.03±0.90 (1.0-5.0) | 3.85±0.82 (1.0-5.0) | 4.01±0.96 (1.0-5.0) | 4.07±0.91 (1.0-5.0) | 4.29±0.87 (1.0-5.0) | 3.85±0.98 (1.0-5.0) | 3.66±0.87 (1.0-5.0) |
| **Average** | 4.12±0.87 (1.0-5.0) | 3.97±0.81 (1.0-5.0) | 4.11±0.92 (1.0-5.0) | 4.15±0.89 (1.0-5.0) | 4.36±0.86 (1.0-5.0) | 3.97±0.96 (1.0-5.0) | 3.75±0.85 (1.0-5.0) |

*Domain 1*, General Satisfaction; *Domain 2*, Technical Quality; *Domain 3*, Interpersonal Manner; *Domain 4*, Communication; *Domain 5*, Financial Aspects; *Domain 6*, Time spent with doctor; *Domain 7*, Accessibility and Convenience.

Similarly, educated populations, particularly those with exposure to superior services and familiarity with stringent quality standards, may harbour elevated expectations, potentially resulting in lower satisfaction levels. However, higher education can also foster a more profound understanding of healthcare system challenges, especially in developing countries, ultimately resulting in better satisfaction. While education level has the potential to influence patient satisfaction in both directions, in the Bhutanese context, our findings demonstrate a negative impact. Similar influences might have affected patient satisfaction across different occupational groups. Bhutan's literacy rate rose from 66.0% in 2017 to 70.2% in 2022 [38]. As the literacy rate improves, patient satisfaction may decline since the level of education appears to affect satisfaction inversely in Bhutan. Additionally, modernization is shifting people from farming and spirituality to modern life-styles, potentially exacerbating this trend. Therefore, Bhutanese healthcare policy and decision-makers should proactively anticipate these changes and implement strategies to address potential challenges in the future.

The satisfaction level is significantly higher among inpatients, older adults, and ever-married groups compared to their counterparts; however, none of these groups are found to be significant predictors of satisfaction. This suggests that belonging to these categories may not significantly increase the likelihood of being satisfied when accounting for other variables. However, differences in analytical methodologies may also contribute to these observed discrepancies. Several prior studies reported varied findings, some aligning with ours, and others conflicting, leading to inconsistent conclusions on patient type, age, and marital status impact on patient satisfaction [20,21,23,26,32,35,39–44].

In our context, it is plausible that inpatients reported higher satisfaction levels due to the provision of more personalised care and supportive environments compared to outpatients. In Bhutan, where elderly individuals are respected and exhibit spiritual tendencies, most experience a good quality of life [45]. This optimistic cultural atmosphere may have positively influenced the satisfaction level in our study. Furthermore, a study conducted in the USA found that physicians are more likely to engage in patient-centric encounters with older patients, who subsequently reported higher satisfaction levels [42]. This interpersonal dynamic might have additionally contributed to higher satisfaction levels among older adults in Bhutan.

Across nations and historical periods, married individuals generally tend to experience greater levels of happiness and satisfaction, although these emotions are subject to the dynamics within their relationships [46–48]. This heightened sense of contentment may extend to their encounters with healthcare services, positively influencing their perceptions and interactions, and ultimately contributing to higher satisfaction levels.

While satisfaction levels do not significantly differ between sexes in our study, females are 1.67 times more likely to be satisfied with healthcare services than males. However, the effect size was small, and statistical power was low for the sex and service utilisation group. This suggests that the study may suffer from inadequate sample size, inherent variability, or very small true differences between the groups. Therefore, future research should address these limitations to confirm the findings and provide more robust conclusions.

The existing literature presents mixed findings regarding the association between sex and patient satisfaction. Some studies report no differences, while others identify one sex as a predictor of higher satisfaction [21,24,32,40,41]. Some researchers suggest that women naturally have lower expectations compared to men, and this difference might have contributed to a greater likelihood of satisfaction among females [21,40]. While it is plausible that innate sex-related characteristics could influence satisfaction, in our context, the presence of healthcare programmes and services tailored for females might also have played a part in the observed higher likelihood of satisfaction.

The insignificant difference among service utilization groups could be due to a consistent delivery of high-standard care by healthcare professionals regardless of visit frequency. Furthermore, the perception formed during the initial encounter might exert a significant influence on subsequent perceptions, leading to consistent satisfaction levels. On the other hand, assessment tools may have lacked the sensitivity to detect subtle differences based on service utilisation frequency. Overall, our study highlights the complex factors shaping patient satisfaction, emphasizing the need for further inquiry to meet the distinct needs of different patient groups.

The MERRH and GCRRH are both state-owned healthcare centres with comparable infrastructure and resources; nevertheless, patient satisfaction is significantly higher at MERRH. This difference might be attributed to the socio-demographic characteristics of the population served. MERRH, located in eastern Bhutan, predominantly serves Sharchop individuals, who are significantly more satisfied than the Lhotshampa individuals primarily served by GCRRH in south-central Bhutan.

Our study shows high overall satisfaction among Bhutanese patients across all seven domains. This finding highlights the concerted efforts of the Bhutanese healthcare system to ensure the provision of quality healthcare services through a comprehensive and systematic approach, including the vigorous implementation of the BHSQA. However, it is important to acknowledge that the surveys were administered by the quality units of the respective healthcare centres, and for illiterate participants, responses were recorded with the assistance of friends or quality assurance officials. This process may have introduced response bias. To mitigate this, healthcare centres could consider implementing online, anonymous survey systems or engaging external evaluators to assess patient satisfaction.

Additionally, Bhutan's culture is deeply influenced by Buddhism in all aspects of life. Buddhist philosophy, which emphasizes compassion, kindness, and non-violence, shapes the values and behaviours of the Bhutanese people. These cultural values profoundly affect how patients perceive and respond to healthcare services. In Bhutan, there is often a tendency to provide positive feedback, even when the service does not fully meet expectations. This may stem from cultural norms that prioritize harmony and respect, which are integral to Buddhist teachings. As a result, patients may focus on expressing gratitude and appreciation for healthcare providers, sometimes refraining from negative feedback to maintain social harmony. These factors warrant careful consideration when interpreting the survey results and evaluating the true extent of patient satisfaction. For optimal service delivery, patients and healthcare professionals must share equal responsibility. In terms of service feedback, patients must provide truthful and reliable feedback, as misleading feedback could hinder opportunities for improvement and undermine the healthcare system. Therefore, in Bhutan, concerned agencies, whether governmental or non-governmental, should promote honesty and accountability in feedback to optimally enhance healthcare services.

The Bhutanese patients are most satisfied in the financial domain, potentially due to Bhutan's provision of free healthcare services. A similar high satisfaction level in the financial domain has also been observed in other studies where healthcare is provided free [23,24]. The constitution of Bhutan mandates the state to provide free access to basic public health services, covering both modern and traditional medicine, for all citizens [49]. Accordingly, the Bhutanese healthcare system provides a broad range of services, including the ex-country referral of complex cases, free of cost. This mode of service delivery, which involves allocating limited financial, infrastructural, and human resources across a broad spectrum of services, can have adverse effects on both service sustainability and quality. While free healthcare services could be currently enhancing patient satisfaction levels in Bhutan, there is a risk of unsustainable expectations and strain on resources. It may induce the perception that all healthcare needs will be met without limits, ultimately impacting patient satisfaction levels.

Given Bhutan's context, establishing dynamic limits of free services is crucial for ensuring sustainable delivery of quality services. Additionally, exploring private participation in delivering high-end services beyond the scope of state-owned centres could serve the public interest more effectively. The domains of general satisfaction, interpersonal manners, and communication also achieved high satisfaction levels, surpassing a rating of four. The inherent friendliness and compassion in Bhutanese society might have contributed to these positive outcomes. Furthermore, a study from Saudi Arabia has identified a positive correlation between financial aspects and other domains, including interpersonal manners and communication [23].

In our study, patients were least satisfied with accessibility and convenience, followed by time spent with doctors and technical quality. Although patient satisfaction varies across countries due to cultural contexts, resource availability, and the effectiveness of the health system [21,25,50,51], studies have frequently reported dissatisfaction in these areas

[24,26–28,39,52,53]. Bhutan has been facing an acute shortage of healthcare professionals for a long period of time. The current doctor-to-population ratio is 0.46 doctors per 1,000 individuals, below the WHO's recommended ratio of 1 per 1,000. Similarly, the nurse-to-population ratio is 2.31 nurses per 1,000, falling short of the global average of 3.7 nurses per 1,000 [54,55]. To improve the specialist healthcare workforce in Bhutan, the Khesar Gyalpo University of Medical Sciences of Bhutan introduced its first Doctorate of Medicine courses in 2014, expanding to nine disciplines [56]. Nonetheless, there remains a significant shortage of specialists in the country. This scarcity might have contributed to lower satisfaction levels in these domains. To address this, policymakers and healthcare academic institutes should consider the expansion of training programmes, offering incentives, strengthening telemedicine, exploring role expansion, improving working conditions, promoting healthcare careers, and establishing a user-friendly online medical appointment system. Notably, in our context, the systematic collection of end-user feedback is limited. As a result, strategies may sometimes lack alignment with the actual needs and expectations of the population. Therefore, piloting an online system for anonymous feedback collection could help policymakers develop strategies that better address the needs of health service users.

While patient satisfaction has demonstrated potential in identifying unmet patient needs and providing valuable insights for quality improvement, other studies have raised concerns about unintended effects associated with satisfaction surveys. These concerns have led to resistance from healthcare professionals, especially regarding their integration with compensation [57,58]. In the US study, patient satisfaction surveys notably decreased physician job satisfaction, prompting some to consider leaving medicine and nearly half to believe it could lead to inappropriate care [59]. Patient satisfaction is crucial for healthcare delivery, yet ensuring job satisfaction and security of healthcare professionals is equally vital. Hence, striking a balance between these factors and exercising caution in utilising patient satisfaction within established limits is imperative.

In Bhutan, where the healthcare quality culture is still evolving, utilising patient satisfaction for CQI is sine qua non. However, policymakers must consider potential unintended consequences if it is used to evaluate healthcare professionals for performance-related decisions. To mitigate potential negative impacts on healthcare professionals, Bhutan could adopt a strategy where satisfaction surveys are used solely for organizational improvement, rather than for individual evaluations or compensation. This approach would allow healthcare providers to prioritise quality improvement without fear of punitive measures, fostering a supportive environment that upholds both patient satisfaction and provider morale.

While secondary data provides valuable insights, it also has several limitations. As a retrospective cross-sectional design, the study cannot capture temporal changes. Recall bias may have influenced the accuracy of participants' responses. Additionally, for illiterate participants, responses recorded by staff or friends could have introduced errors or positive bias. Response bias, driven by factors such as literacy levels, cultural norms, or social desirability, may have influenced how participants understood and answered the survey questions. Although the survey captured all major ethnic groups in Bhutan, the proportions of respondents from each group may not align with their representation in the broader population, potentially limiting the generalizability of the findings.

## Conclusions

Our study provides valuable insights into patient satisfaction levels and their predictors in Bhutan. We found that ethnicity, occupation, and education level were significant predictors of satisfaction. Specifically, Sharchop and other ethnic groups, farmers, religious personnel, and other occupational groups, as well as illiterate groups exhibited significantly higher satisfaction levels and were more likely to be satisfied with healthcare services. With the anticipated changes in certain socio-demographic characteristics of the Bhutanese population, patient satisfaction is likely to decline. Therefore, healthcare policy and decision-makers should implement targeted interventions to address these shifts.

The overall patient satisfaction among Bhutanese patients is high, with a score of 4.06 on a 5-point Likert scale. Bhutanese patients reported the highest satisfaction in the financial domain, reflecting the state's success

in providing free healthcare services. Nonetheless, areas such as accessibility and convenience, time spent with doctors, and technical quality require improvement to enhance patient satisfaction. To address these challenges, implementing dynamic limits on free services, encouraging private participation in advanced services beyond the scope of state-owned centres, strengthening the healthcare workforce, and embracing innovative approaches can help ensure sustainability and improve service quality. As Bhutan's healthcare system evolves, patient satisfaction data should be systematically integrated into CQI efforts. Regular nationwide assessments must be implemented to identify areas for improvement while ensuring a balanced approach to safeguard healthcare professionals' morale and care standards.

While our findings provide valuable insights, further research is necessary to identify the factors driving differences in patient satisfaction. A mixed-methods approach could uncover key contributors to these variations. Longitudinal and comparative studies across multiple hospitals could track shifts in patient expectations, while ethnographic research could reveal cultural and systemic influences. These findings could inform evidence-based policies to better address patient needs.

## Acknowledgments

We are thankful to the survey respondents for their participation and valuable feedback. Additionally, we are grateful to the hospital management of MERRH and GCRRH for their permission to access the archived patient satisfaction data.

## Author contributions

**Conceptualization:** Kuenzang Dorji, Kinga Jamphel, Jigme Kelzang, Ugyen Pelmo, Hem Kumar Nepal, Sonam Zangpo, Sonam Wangdi, Karma Galey.

**Data curation:** Kuenzang Dorji, Kinga Jamphel, Jigme Kelzang, Ugyen Pelmo, Hem Kumar Nepal, Sonam Zangpo, Sonam Wangdi, Karma Galey.

**Formal analysis:** Kuenzang Dorji, Kinga Jamphel, Jigme Kelzang, Ugyen Pelmo, Hem Kumar Nepal, Sonam Zangpo, Sonam Wangdi, Karma Galey.

**Investigation:** Kuenzang Dorji, Kinga Jamphel, Jigme Kelzang, Ugyen Pelmo, Hem Kumar Nepal, Sonam Zangpo, Sonam Wangdi, Karma Galey.

**Methodology:** Kuenzang Dorji, Kinga Jamphel, Jigme Kelzang, Ugyen Pelmo, Hem Kumar Nepal, Sonam Zangpo, Sonam Wangdi, Karma Galey.

**Project administration:** Kuenzang Dorji, Kinga Jamphel, Jigme Kelzang, Ugyen Pelmo, Hem Kumar Nepal, Sonam Zangpo, Sonam Wangdi, Karma Galey.

**Resources:** Kuenzang Dorji, Kinga Jamphel, Jigme Kelzang, Karma Galey.

**Software:** Kuenzang Dorji, Kinga Jamphel, Jigme Kelzang, Karma Galey.

**Supervision:** Kuenzang Dorji, Kinga Jamphel, Jigme Kelzang, Karma Galey.

**Validation:** Kuenzang Dorji, Kinga Jamphel, Jigme Kelzang, Ugyen Pelmo, Hem Kumar Nepal, Sonam Zangpo, Sonam Wangdi, Karma Galey.

**Visualization:** Kuenzang Dorji, Kinga Jamphel, Jigme Kelzang, Ugyen Pelmo, Hem Kumar Nepal, Sonam Zangpo, Sonam Wangdi, Karma Galey.

**Writing – original draft:** Kuenzang Dorji, Kinga Jamphel, Jigme Kelzang, Karma Galey.

**Writing – review & editing:** Kuenzang Dorji, Kinga Jamphel, Jigme Kelzang, Ugyen Pelmo, Hem Kumar Nepal, Sonam Zangpo, Sonam Wangdi, Karma Galey.

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
