## [Decision Letter · Decision Letter 0]

PONE-D-24-44865Patient satisfaction: Insights from the regional referral hospitals in BhutanPLOS ONE

Dear Dr. Dorji,

Thank you for submitting your manuscript to PLOS ONE. After careful consideration, we feel that it has merit but does not fully meet PLOS ONE’s publication criteria as it currently stands. Therefore, we invite you to submit a revised version of the manuscript that addresses the points raised during the review process.

**ACADEMIC EDITOR: **Please address the comments of the 7 reviewers. Thank you.

We look forward to receiving your revised manuscript.

Kind regards,

Ian Christopher N Rocha, MD, MBA, MHSS

Academic Editor

PLOS ONE

Journal Requirements:

Additional Editor Comments:

Please address the comments of the 7 reviewers. Thank you.

Reviewers' comments:

Reviewer's Responses to Questions

**Comments to the Author**

1. Is the manuscript technically sound, and do the data support the conclusions?

Reviewer #1: No

Reviewer #2: Yes

Reviewer #3: Partly

Reviewer #4: Yes

Reviewer #5: Yes

Reviewer #6: Yes

Reviewer #7: Yes

2. Has the statistical analysis been performed appropriately and rigorously? 

Reviewer #1: No

Reviewer #2: Yes

Reviewer #3: Yes

Reviewer #4: Yes

Reviewer #5: Yes

Reviewer #6: Yes

Reviewer #7: Yes

3. Have the authors made all data underlying the findings in their manuscript fully available?

Reviewer #1: Yes

Reviewer #2: No

Reviewer #3: No

Reviewer #4: No

Reviewer #5: No

Reviewer #6: Yes

Reviewer #7: No

4. Is the manuscript presented in an intelligible fashion and written in standard English?

Reviewer #1: Yes

Reviewer #2: Yes

Reviewer #3: No

Reviewer #4: Yes

Reviewer #5: Yes

Reviewer #6: Yes

Reviewer #7: Yes

5. Review Comments to the Author

Reviewer #1: Study title:

You need to re-write study title as follows

“Evaluation of patient satisfaction in the regional referral hospitals in Bhutan: a cross-sectional study”.

Abstract:

You need to re-write the abstract in a standard format including the headings ‘Background, Methods, Results, and conclusion’.

Main text:

Introduction:

It is well-written and appreciated.

Materials and methods:

You need to re-write methods section including the following sub-headings as a separate entity:

-study design

-study site

-study duration

-study setting

-study population

-inclusion criteria

-exclusion criteria

-sample size determination

-sampling method

-study variables

-study procedures

-data management

-data analysis

Results:

In Table 1, you can compare the socio-demographic variables and clinical profiles of the respondents between Gelephu and Mongar hospital, and that will give more meaning to the table.

You have inappropriately used one-way ANOVA analysis. ANOVA can be used only if you have more than 2 groups to compare quantitative variables. So, you can delete current table 2, and construct new table 2 to compare patient satisfaction scores across seven domains between Gelephu and Mongar hospital.

You need to re-write table 3 to include univariate analysis in one column and multivariate logistic regression analysis in another column. This will give a better understanding to the reader about the factors associated with poor patient satisfaction or well satisfied patient.

Table 4 doesn’t give any scientific inferences. You can delete table 4.

Reviewer #2: Abstract

1. Rather than saying "archived patient satisfaction data," you could clarify the source or specific nature of this data.

Introduction

The introduction provides a solid contextual foundation for the study. Minor stylistic adjustments would improve readability and conciseness, particularly by condensing and refining some sentences.

1. The phrase “meeting patient’s needs is imperative” could be made more specific, and the sentence starting “Starting from the early 1980s” might benefit from a more concise phrasing.

2. Also, the lack of a universally accepted definition of patient satisfaction could be condensed to focus more on its role as a quality measure.

3. The phrase “meticulously developed tool” to describe the PSQ-18 may appear overly emphatic in a scientific context.

Materials and methods

1. Confirm whether the survey sites were called “MERRH” or “ERRH” and “GCRRH” or “CRRH”.

2. Consider adding a brief statement on the justification for modifying specific terminology in the PSQ-18 and whether there was any assessment of the adapted tool’s reliability or validity in this new context.

3. Add a sentence specifying the handling of negative items during re-scaling, as this will aid replicability and ensure clarity in interpretation.

4. Briefly explain what the Dunnett grouping is, why the Dunnett grouping was used to identify distinct groups and why a significance level of 0.005 was selected. The significance level of 0.005 is strict and unusual (more commonly 0.05), so a rationale for this choice would also improve clarity.

Result

1. Satisfaction scores vary significantly by age, with older respondents generally reporting higher satisfaction levels. This finding could be influenced by generational differences in expectations of healthcare, which might warrant a brief discussion of the interpretation.

2. The study shows a predominant representation of Sharchop ethnic group respondents (57.05%), and satisfaction scores are also the highest among them. Given Bhutan's ethnic diversity, exploring whether this reflects the general population distribution or biases in sampling might be helpful.

3. Illiterate individuals reported higher satisfaction scores than those with higher education levels, which might suggest varying expectations based on education. This could be discussed as it may reveal important insights into patient perceptions of healthcare quality across different educational backgrounds.

4. The choice of control groups (e.g., using the 18-44 age group or married status as the control) appears sound. However, mentioning why these groups were selected as the baseline for comparison could strengthen interpretability.

5. Given that significant differences in satisfaction levels were observed between MERRH and GCRRH, it may be beneficial to explore specific differences in facilities, staff interaction, or procedural efficiency between the two. These insights could further contextualize why one center reports higher satisfaction levels.

Discussion

1. The authors astutely highlight Buddhism's influence on Bhutanese healthcare interactions, leading to positive patient responses. However, there may be potential biases, as indicated, that result from cultural expectations of politeness and contentment, particularly when surveys are administered by quality units. It may be valuable to propose alternative assessment methods, such as third-party evaluators or anonymous surveys, to better gauge satisfaction independently of cultural expectations.

2. The potential for satisfaction surveys to affect provider performance and motivation is a valid concern. The authors might expand this discussion by considering how Bhutan could develop a model where satisfaction surveys are used solely for improvement and not linked to individual compensation or evaluations, thus mitigating potential professional dissatisfaction and unintended care implications.

Conclusion

1. The authors could suggest areas for further research, such as longitudinal studies or comparative studies with similar healthcare systems in culturally similar countries. This would aid policymakers in anticipating and addressing shifts in patient expectations and satisfaction.

Recommendations

1. If the paper could discuss the limitations explicitly if any would strengthen the conclusions.

2. Qualitative follow-up with respondents could add depth to the quantitative findings, helping to contextualize why specific groups, such as the illiterate population, report higher satisfaction.

Reviewer #3: Abstract:

Good abstract

L32-33: you may just mention one test

L36- mention all the ethnic groups which are significant not just others, significant by how much?

L35-41: please mention the findings in figures (% or numbers, etc) in brackets.

L42-46: rephrase it and make it clear, seems sentences are diluting and repeating.

L47-49: What makes you conclude this, it should be related to your research findings not just hypothetical talk, what do you mean by high-end services, there was no finding on healthcare workforce but how come you conclude on this. Please be very specific, just mention what you get from your research.

Introduction:

General comment- keep citation number in sequence, and so does write the flow accordingly.

L87-88: rephrase the sentence to make more urging need than just saying not utlized.

L88-90: are you sure this hampers the healthcare delivery service, if yes please support by evidence.

Materials and Methods:

L94-98: Why only Mongar and Gelephu hospital, mention what is the reason for choosing these two hospitals. Why JDWNRH national referral hospital was not selected? Data says more than 60% of country's patient visit this hospital, I guess you have lost good amount of data.

L98-99: procedure is not clear, how the patient is being enrolled. Please mention in between these two lines. Showing in flowchart would be great though.

L125-126: I think you dont have to mention about the waiver for consent, this is feedback survey anyways.

RESULTS:

L146- are you sure about the clinical variables, I cant see in the table.

L148-166: Just keep few important ones, no need to describe all the variables in sentence, tables speaks.

L167-168: no need to mention

Table 4: is not good table, lot of data into it and it speaks less about it. You may delete the table or modify it. Define and show the satisfaction level, how much is dissatisfied, which score is moderate and which score is satisfied, please mention either in method section or along with the table 4.

DISCUSSION:

General comment- need to refine it more clearly. Be specific with your results and support it with previous studies.

L194-197: rephrase the sentence, i dont think so, please support by findings, what is the proportion of the patient seen and the population of these two region when you say it represents.

L213-215: How?? need more data to support

Please mention about limitations and the strength of the study before conclusion.

CONCLUSION:

L334-341: no need to reflect, those are less significant to mention

Please mention the overall satisfaction level of the patients in the conclusion, so that readers take away the message.

Reviewer #4: Thank you for the opportunity to review this work on patient satisfaction, which is quite a young but very important indicator of quality healthcare services.

Here are my comments;

Abstract

Indicate the level of patient satisfaction as found in your study

Consider including p-values for variables that are statistically significantly associated with patient satisfaction.

Introduction.

Strengths. The introduction is well structured with relevant literature capturing the concept of patient satisfaction. The authors clearly indicate the different factors that influence satisfaction and they also note that there is scarcity of published literature on the subject in Bhutan.

Criticism & suggestions. The researchers need to clearly indicate the role of patient satisfaction in healthcare highlighting the potential contribution of the study to practice. In lines 81-86, authors mention about the transition towards quality oriented care in their setting. More literature to capture the interventions from the Bhutan authorities implemented to improve patient satisfaction should be added. Authors should also add more evidence regarding patient satisfaction from other settings especially preceding the description on the factors influencing satisfaction.

What are different domains of satisfaction assessed in other settings and how do these assessment link into the assessments in Bhutan?

Materials and methods

The current structure of the methods section does not allow for smooth transition by the reader. The authors should consider structuring the section into subsections such as study design, study population, study setting, data collection tools, data collection procedures etc....

The targeted population in this study is not clear.

Was the data collected secondary or primary? Lines 99-100 seem to indicate the latter but from previous statements, one could infer that the data was secondary. This needs to be clarified in your writeup. If this was secondary data, discuss the possible biases.

The authors need to indicate the time points/periods from which the data was obtained. Do you think this affects satisfaction?

If this was secondary data, did the authors design a data abstraction? Was this informed by the PSQ18? Has the PSQ18 been utilized in other research studies? Is it a valid and reliable tool for establishing patient satisfaction? What was the internal consistency in this current study? The researchers should provide the tool used in this assessment as a supplementary material for replicability of the study in other settings.

Line 177 describes how the researchers quantified patient satisfaction by getting the average scores across 7 domains of the tool. Is this the standard approach or this was adapted?

Results

Presenting the results with subsections as per the meaning of the findings would improve the flow of information in this section.

Author need to revise the use of some words such as control group, comparison group as these could mean different parameters in different study designs.

Were the odds ratios presented in table 3 crude or adjusted? You need to consider including both.

According to the authors, the variables ethnic group and occupation were significantly associated with satisfaction. What was the rationale for creating the categories under these two variables? Are the individuals under the same group having similar characterization? and if not, could it be a biased assignment? For instance, in the category of occupation, why are students in the same group as civil servants?

Based on your analysis plan, you need to indicate in the results section the proportion of those patients who are satisfied and dissatisfied.

Discussion

The researchers have done a great job in discussing and linking their findings to the already existing body of knowledge and the implications of the findings are clear. This section can be improved by;

Providing a detailed discussion of the limitations associated with methodological weaknesses, highlighting their impact on the generalizability of the study findings.

The structuring of the discussion should be changed. The level of satisfaction should be discussed first followed by the factors associated with satisfaction.

Reviewer #5: I feel that the paper is well written and its findings are very significant to health policy maker and also for the country, this study will serve as a baseline on patient satisfaction level for future researches.

Reviewer #6: This manuscript presents a timely and relevant study on patient satisfaction in Bhutan, a country with a developing healthcare system. The study's findings, particularly the identification of ethnicity, occupation, and education level as significant predictors of patient satisfaction, offer valuable insights for healthcare policymakers and professionals in Bhutan and other countries with similar healthcare systems.

Strengths

The research addresses the important question of patient satisfaction within the Bhutanese healthcare system, which is undergoing a transition towards a quality-oriented culture. This is particularly relevant given the global emphasis on patient-centered care and the need to understand the factors that influence patient satisfaction in diverse healthcare settings.

The study adds to the existing body of literature by providing empirical evidence on patient satisfaction within the specific context of Bhutan. This is a valuable contribution as there is limited research on this topic in Bhutan, and the study's findings offer unique insights into the factors that influence patient satisfaction in a developing healthcare system

The conclusions are consistent with the evidence and arguments presented in the study. The authors have accurately summarized the key findings and their implications for healthcare policy and practice

Areas for Improvement

The authors mention the influence of Bhutan's cultural context on patient satisfaction, they could expand this discussion to provide a more nuanced understanding of how cultural factors, such as the emphasis on compassion and the role of Buddhism, may influence patient expectations and perceptions of healthcare services.

“This study retrospectively analysed patient satisfaction survey responses collected from Mongar Eastern Regional Referral Hospital (MERRH) and Gelephu Central Regional Referral Hospital (GCRRH). The routine paper-based surveys were conducted throughout April 2024.” The manuscript lacks a detailed description of the data collection process, including the sampling method. Providing this information would strengthen the study's methodology and enhance the credibility of the findings

The manuscript could benefit from a more specific and actionable recommendations for healthcare policymakers and professionals. For instance, the authors could provide concrete examples of targeted interventions to address the anticipated decline in patient satisfaction due to changing socio-demographic characteristics.

Writing Quality

The paper is well-written, with clear and easy-to-read text. The authors have used appropriate language and style for an academic audience.

Reviewer #7: General Comments

Dorji K et al. attempted to provide valuable insights into patient satisfaction levels at two referral hospitals in Bhutan, using retrospective data and exploring socio-demographic predictors of satisfaction. This research addresses an important gap in understanding healthcare quality in Bhutan. However, there are areas where further clarity would strengthen the findings. Enhancing transparency in the methodology, particularly around sample size, sampling methods, and survey administration, would strengthen this study. Addressing potential biases and improving the contextual richness of the discussion would also benefit readers and support the study’s practical implications for healthcare policy in Bhutan.

Abstract

1. Specificity in reporting results (Lines 22-49): The abstract provides a clear summary but could benefit from additional specificity. Including exact figures for satisfaction levels or effect sizes of significant predictors would make the results more informative for the reader.

Introduction

1. Bhutan’s health system (Lines 69-70): While the introduction discusses patient satisfaction broadly, it could be enhanced by offering more context on Bhutan’s healthcare system. For e.g., more emphasis on free healthcare, and the challenges associated with rural accessibility and workforce shortages would provide readers with a clearer understanding of why patient satisfaction is critical in this setting.

2. Cite BHSQA (Lines 75-76)

Materials and Methods

1. Study design and sampling method (Lines 93-128): Clarify the sampling method, especially whether it was a convenience sample or if random sampling was applied. Sampling methods are essential to understanding potential biases, as convenience sampling can limit generalizability. The use of a retrospective cross-sectional design is appropriate for the data available, but its limitations should be noted, as this design cannot capture temporal changes in satisfaction or causality. Future studies could benefit from a longitudinal approach to observe changes in satisfaction over time, especially with healthcare reforms.

2. Inclusion of western region hospitals (Lines 93-128): The study’s sample is limited to hospitals in eastern and southern Bhutan, which may not capture the full demographic diversity of the population. Including a referral hospital from the western region, such as JDWNRH, would enhance representativeness, as patients in the west may have different expectations and healthcare perceptions.

3. Mode of survey administration (Lines 99-100): If health professionals conducted surveys for illiterate patients, this could introduce interviewer bias, leading patients to feel pressured to respond positively. The manuscript should clarify who administered the surveys and acknowledge any potential interviewer effects. A more consistent survey administration mode, such as anonymous or third-party administration, could reduce response bias, especially among illiterate respondents.

4. Sample size (Lines 93-128): The study does not provide a power analysis or sample size calculation. Although 915 respondents seem more than adequate, a statistical justification would add methodological correctness. Reporting a power analysis based on effect sizes would strengthen the credibility of the sample size in capturing true effects across socio-demographic groups.

5. The method could benefit from greater transparency regarding the survey modification process for PSQ-18. Provide additional information on the survey's adaptation process and provide the modified PSQ-18 as supplementary file (Lines 100-107).

6. Multiple comparisons correction in ANOVA (Lines 114-115): The authors used one-way ANOVA with Dunnett grouping for multiple comparisons across socio-demographic groups, but it lacks a correction for multiple comparisons. Given the numerous factors compared, using a method like Bonferroni or Holm correction would reduce the likelihood of Type I error. This correction would ensure that significant differences between groups are robust and not due to chance alone.

7. Significance levels (Line 121): The manuscript uses a significance level of p < 0.005, which is stricter than the conventional p < 0.05. This could strengthen findings by reducing the risk of Type I errors, but it would be good to clarify why this threshold was chosen. If a multiple-comparisons correction (such as the Bonferroni correction) was applied to account for repeated testing across multiple domains, mentioning this would add more detail.

Results

1. Visual summaries (Lines 129-187): While the tables are comprehensive, adding visualizations (e.g., bar charts or histograms) for key socio-demographic variables and satisfaction levels across domains would improve accessibility and make trends easier to interpret. These visuals could also help in identifying outliers or skewness that might affect parametric assumptions in ANOVA.

2. Wide confidence intervals in logistic regression (Lines 171-175): Some odds ratios show wide confidence intervals, indicating limited precision, possibly due to small subgroup sizes or inherent variability. Briefly acknowledging this limitation and its potential impact on the reliability of estimates would add transparency to the findings.

Discussion

1. Cultural factors and contextual influences on satisfaction: The discussion effectively ties findings to existing literature but could benefit from a more in-depth look at Bhutan-specific cultural factors that might influence satisfaction. For instance, the cultural respect for authority figures and the potential for social desirability bias in survey responses could influence satisfaction ratings, particularly among illiterate respondents.

2. Explanation of higher satisfaction among illiterate participants (Lines 199-218): The significantly higher satisfaction among illiterate patients is noteworthy but could partly reflect interviewer influence if surveys were administered by healthcare professionals. Discussing this potential interviewer effect here would provide insight into how survey mode might impact satisfaction scores across education levels.

3. Limitations in regional representation (Lines 188-327): The findings from Mongar ERRH and Gelephu CRRH may not fully represent national patient satisfaction trends in Bhutan, as the absence of data from the west (e.g., Thimphu) could exclude urban patient perspectives and include more representation of ngalops. Acknowledge this limitation in the discussion and suggest that future research include hospitals from multiple regions for a more balanced national representation.

4. Policy implications and recommendations (Lines 278-350): The authors have provided valuable recommendations for improving patient satisfaction including enhancing accessibility and convenience. Strengthening this section with specific, immediate actions, for e.g., implementing a pilot program for anonymous feedback collection, could make the policy implications more actionable for health administrators in Bhutan.

6. PLOS authors have the option to publish the peer review history of their article (what does this mean?). If published, this will include your full peer review and any attached files.

Reviewer #1: No

Reviewer #2: **Yes: **Kinley Gyem

Reviewer #3: No

Reviewer #4: **Yes: **Businge Alinaitwe

Reviewer #5: No

Reviewer #6: No

Reviewer #7: No

---

## [Author Response · Author response to Decision Letter 1]

23 Dec 2024

Response to Reviewers

Reviewer #1

Study title:

Comment: You need to re-write the study title as follows

“Evaluation of patient satisfaction in the regional referral hospitals in Bhutan: a cross-sectional study”.

Response: Thank you for the comment. We have made the changes as follow:

“Patient satisfaction in regional referral hospitals of Bhutan: Insights from a Cross-Sectional Study”

Abstract:

Comment: You need to re-write the abstract in a standard format including the headings ‘Background, Methods, Results, and Conclusion’.

Response: Thank you for your valuable feedback. Initially, we provided the abstract in an unstructured format, consistent with the published articles in PLOS ONE, where abstract formatting varies. However, in light of your comment, we have revised the abstract into a structured format, aligning it with the conventions of structured abstracts for clarity and comprehensiveness.

Main text:

Introduction:

Comment: It is well-written and appreciated.

Response: Not required

Materials and methods:

Comment: You need to re-write methods section including the following sub-headings as a separate entity:

-study design

-study site

-study duration

-study setting

-study population

-inclusion criteria

-exclusion criteria

-sample size determination

-sampling method

-study variables

-study procedures

-data management

-data analysis

Response: Dividing the Materials and Methods section into individual sub-headings as recommended would result in a very brief and fragmented write-up, which might not effectively present the detailed methodology of the study. To ensure a more cohesive and comprehensive narrative, we have grouped related sub-headings together into four broader categories, each addressing a key aspect of the study. This approach allows us to present the methodology concisely while maintaining the clarity and flow necessary for readers to understand the process.

The grouped sub-headings are as follows:

1. Study design and setting

2. Study population and survey instrument

3. Data management and analysis

4. Ethical considerations

Results:

Comment: In Table 1, you can compare the socio-demographic variables and clinical profiles of the respondents between Gelephu and Mongar Hospital, and that will give more meaning to the table.

Response: We sincerely thank the reviewer for their insightful comment. We agree that presenting the socio-demographic variables and clinical profiles of the respondents from both hospitals provides valuable additional insights. Accordingly, we have amended Table 1 to include these details, enabling a more meaningful comparison.

Comment: You have inappropriately used one-way ANOVA analysis. ANOVA can be used only if you have more than 2 groups to compare quantitative variables. So, you can delete current table 2, and construct new table 2 to compare patient satisfaction scores across seven domains between Gelephu and Mongar hospital.

Response: We sincerely thank the reviewer for their observation regarding the application of one-way ANOVA. We would like to clarify that while ANOVA is commonly used to compare more than two groups, it is also valid for comparing two groups. In fact, when there are exactly two groups, the results of a one-way ANOVA are mathematically equivalent to those obtained from a t-test. In our study, we applied one-way ANOVA consistently across all variables for uniformity, as some variables involve more than two groups (e.g., Age, Ethnic Group, and Marital Status).

To adhere to general practice, we have now used the 2-sample t-test for variables such as Age, Sex, and Service Utilisation, as these have only two groups. To clarify the statistical methods used, a footer has been added to the table specifying where the one-way ANOVA and t-test have been performed. However, there have been no changes in the significance level.

We would like to express our sincere thanks for your insightful suggestion to compare patient satisfaction across the seven domains between the two hospitals. We agree that comparing patient satisfaction across the seven domains between the two hospitals is a valid approach to analyzing the data. However, as we have already provided an overall comparison of patient satisfaction between Gelephu and Mongar hospitals in table 2, we believe that focusing on patients' socio-demographic and clinical variables offers more meaningful insights for actionable improvements at a national level. Both hospitals are state-owned, with comparable services and infrastructure, making them suitable for broader comparisons.

Rather than comparing hospital-to-hospital satisfaction across the domains, we chose to examine factors such as age, education, and occupation, as these are critical in identifying specific groups that may benefit from targeted interventions. This approach aligns with the goal of improving patient satisfaction more effectively across the country, addressing the needs of diverse population groups, and guiding healthcare policies.

While we understand and acknowledge the value of direct hospital comparisons, we firmly believe that examining the socio-demographic and clinical variables will provide a more comprehensive understanding of the factors driving patient satisfaction, with direct implications for policy and practice at a national level.

Comment: You need to re-write table 3 to include univariate analysis in one column and multivariate logistic regression analysis in another column. This will give a better understanding to the reader about the factors associated with poor patient satisfaction or well-satisfied patient.

Response: We sincerely thank the reviewer for their insightful feedback and suggestion to include univariate analysis alongside multivariate logistic regression analysis. Patient satisfaction is a complex outcome influenced by multiple interconnected factors. Univariate analysis, while useful for initial explorations, might provide misleading information by failing to account for confounding variables. For example, a variable may appear to have a significant association with patient satisfaction in univariate analysis but lose its significance when adjusted for other factors. Conversely, some associations may only emerge after adjusting for confounders.

Given the multifactorial nature of patient satisfaction, we believe that presenting adjusted odds ratios derived from multivariate logistic regression provides a more accurate and meaningful understanding of the factors associated with satisfaction. To enhance clarity, we have amended the table to explicitly label the odds ratios as "Adjusted Odds Ratios"

Comment: Table 4 doesn’t give any scientific inferences. You can delete table 4.

Response: Thank you for your feedback. Table 4 provides a detailed breakdown of satisfaction levels across different socio-demographic groups and domains of care. This information is crucial for identifying specific areas that require improvement. In our experience, such descriptive details are essential for understanding the nuances of patient satisfaction, especially when striving to enhance the quality and safety of healthcare service delivery. By presenting these details, we aim to offer insights that can guide targeted interventions, ensuring that improvements are aligned with the needs of different patient groups. We believe that these specifics are key to informed decision-making and evidence-based policy formulation.

Reviewer #2

Abstract

Comment: Rather than saying "archived patient satisfaction data," you could clarify the source or specific nature of this data.

Response: To address the reviewer's comment, the section in the abstract has been revised as follows:

“We conducted a retrospective analysis of patient satisfaction survey responses archived in the quality assurance unit of two tertiary healthcare centres in Bhutan: Mongar Eastern Regional Referral Hospital and Gelephu Central Regional Referral Hospital. The routine surveys, administered throughout April 2024, utilised an adapted version of the Patient Satisfaction Questionnaire-18. The data were analysed using descriptive and inferential statistics.”

Introduction

The introduction provides a solid contextual foundation for the study. Minor stylistic adjustments would improve readability and conciseness, particularly by condensing and refining some sentences.

Comment: The phrase “meeting patient’s needs is imperative” could be made more specific, and the sentence starting “Starting from the early 1980s” might benefit from a more concise phrasing.

Response: Thank you for the suggestion. We chose to keep the phrase "meeting patients' needs is imperative" broad because patient needs are inherently diverse and encompass various dimensions (e.g., clinical, emotional, psychological, etc.). Listing all potential needs would risk being overly exhaustive and detract from the statement's clarity.

Regarding the sentence starting with “Starting from the early 1980s,” we agree that conciseness can enhance readability. The revised section now reads:

“Since the early 1980s, there have been ongoing efforts to comprehend and measure patient satisfaction, and gradually, it has been acknowledged as a crucial component in delivering quality healthcare.”

Comment: Also, the lack of a universally accepted definition of patient satisfaction could be condensed to focus more on its role as a quality measure.

Response: Throughout the document, we have emphasized the role of patient satisfaction as a measure of quality. While we agree that the focus should remain on its positive impact, we also believe it is essential to acknowledge its inherent ambiguity. Therefore, we felt it necessary to mention the lack of a universally accepted definition, which reflects this ambiguity. Additionally, the sentence in question is brief, consisting of only 14 words, and we believe it does not detract from the overall clarity of the manuscript.

Comment: The phrase “meticulously developed tool” to describe the PSQ-18 may appear overly emphatic in a scientific context.

Response: Thank you for the feedback. We agree that the term 'meticulously developed tool' might be overly emphatic in a scientific context. We have revised the sentence to ensure a more neutral tone:

“To gauge patient satisfaction, Bhutanese healthcare centres utilize the Patient Satisfaction Questionnaire 18 (PSQ-18), a well-validated tool noted for its brevity and efficacy across various contexts.”

Materials and methods

Comment: Confirm whether the survey sites were called “MERRH” or “ERRH” and “GCRRH” or “CRRH”.

Response: Thank you for the feedback. Currently, at the ministry level, there is no consensus on the official terminology, except for the use of "regional referral hospital." Both in official and non-official correspondences, abbreviations such as MERRH/ERRH/MRRH and GCRRH/CRRH/GRRH have been used interchangeably.

According to the official websites of the hospitals, Mongar is referred to as the Mongar Regional Referral Hospital (https://www.mrrh.gov.bt/), while Gelephu is called the Central Regional Referral Hospital (http://www.sarpang.gov.bt/institution/central-regional-referral-hospital), further highlighting this inconsistency. Moreover, an enquiry with the hospital administration confirmed the lack of a standardized term and no official confirmation from the ministry. In light of this, we believe there is no need to change the abbreviations of the study sites at this time.

Comment: Consider adding a brief statement on the justification for modifying specific terminology in the PSQ-18 and whether there was any assessment of the adapted tool’s reliability or validity in this new context.

Response: Thank you for the feedback. The modifications made to the PSQ-18 in the hospital’s survey are detailed in the Methodology section. There were no major changes to the original questionnaire; the only adjustments were the replacement of the terms "doctor" and "doctor’s office" with "healthcare professionals" and "healthcare centre," respectively. These modifications were made to ensure the tool's broader applicability across various healthcare settings while preserving the original intent of the questions. To prevent any confusion, we have replaced the term "modified version" with "adapted version," as the changes did not alter the original structure or purpose of the tool.

As this is a retrospective study, there is limited scope for conducting a formal assessment of the tool's reliability or validity in this new context. Given that the adaptations did not impact the core framework of the PSQ-18, no additional reliability or validity testing was performed.

Comment: Add a sentence specifying the handling of negative items during re-scaling, as this will aid replicability and ensure clarity in interpretation.

Response: Thank you for your comment. The handling of negative items during re-scaling has been outlined in the Methodology section. We followed the standard re-scaling procedure for the PSQ-18 negative items, as recommended in the published guidelines, and applied it consistently in our analysis. Since this is a well-established practice, we did not feel it necessary to provide an exhaustive description in the methodology. For further reference, the published article detailing this procedure can be accessed via the following link: https://www.rand.org/content/dam/rand/www/external/health/surveys_tools/psq/psq18_scoring.pdf

Comment: Briefly explain what the Dunnett grouping is, why the Dunnett grouping was used to identify distinct groups and why a significance level of 0.005 was selected. The significance level of 0.005 is strict and unusual (more commonly 0.05), so a rationale for this choice would also improve clarity.

Response: Performing one-way ANOVA on Minitab provides the following grouping information:

1. Tukey

2. Fisher

3. Dunnett

4. Hsu MCB

5. Games-Howell

Although all the aforementioned grouping tests were performed, only the results of Dunnett's comparison were reported in the manuscript. The findings from all methods were consistent, with no significant differences observed. It is generally recommended to focus on specific group comparisons rather than conducting an overall comparison, if possible, as these methods tend to be more powerful. For example, Tukey's method generally provides less precise confidence intervals and less powerful hypothesis tests compared to Dunnett's or Hsu’s MCB.

In our study, since we already have a mean satisfaction level for each subgroup, we opted to use either the lowest or highest score as a reference point. This approach allowed for more precise and powerful comparisons, aligning to enhance statistical accuracy in our analysis.

Thank you for noting the significance level. Due to a typographical error, we initially mentioned a significance level of 0.005; however, all analyses were conducted at a significance level of 0.05. The correction has been applied.

Result

Comment: Satisfaction scores vary significantly by age, with older respondents generally reporting higher satisfaction levels. This finding could be influenced by generational differences in expectations of healthcare, which might warrant a brief discussion of the interpretation.

Response: In the Results section, we have presented only the findings, in accordance with standard practice. Brief discussions of these findings are provided in the Discussion section, specifically between lines 229 and 235. Additionally, in lines 213 to 215, we have discussed the rising literacy rate and its potential negative impact on patient satisfaction. The generational difference in healthcare expectations can be significantly influenced by literacy levels, and this has been addressed accordingly.

Comment: The study shows a predominant representation of Sharchop ethnic group respondents (57.05%), and satisfaction scores are also the highest among them. Given Bhutan's ethnic diversity, exploring whether this reflects the general population distribution or biases in sampling might be helpful.

Response: Thank you for your comment. As noted in the published literature (e.g., The Kingdom of Bhutan Health System Review), there is no available data on the precis

---

## [Decision Letter · Decision Letter 1]

PONE-D-24-44865R1Patient satisfaction in regional referral hospitals of Bhutan: Insights from a cross-sectional study.PLOS ONE

Dear Dr. Dorji,

Thank you for submitting your manuscript to PLOS ONE. After careful consideration, we feel that it has merit but does not fully meet PLOS ONE’s publication criteria as it currently stands. Therefore, we invite you to submit a revised version of the manuscript that addresses the points raised during the review process.

**ACADEMIC EDITOR: **Please address the remaining minor comments of the reviewers.

We look forward to receiving your revised manuscript.

Kind regards,

Ian Christopher N Rocha, MD, MBA, MHSS

Academic Editor

PLOS ONE

Journal Requirements:

Additional Editor Comments:

Please address the comments of the 9 reviewers.

Reviewers' comments:

Reviewer's Responses to Questions

**Comments to the Author**

1. If the authors have adequately addressed your comments raised in a previous round of review and you feel that this manuscript is now acceptable for publication, you may indicate that here to bypass the “Comments to the Author” section, enter your conflict of interest statement in the “Confidential to Editor” section, and submit your "Accept" recommendation.

Reviewer #2: All comments have been addressed

Reviewer #3: All comments have been addressed

Reviewer #4: All comments have been addressed

Reviewer #5: All comments have been addressed

Reviewer #7: All comments have been addressed

Reviewer #8: All comments have been addressed

Reviewer #9: All comments have been addressed

2. Is the manuscript technically sound, and do the data support the conclusions?

Reviewer #2: Yes

Reviewer #3: Partly

Reviewer #4: (No Response)

Reviewer #5: Yes

Reviewer #7: Yes

Reviewer #8: Yes

Reviewer #9: Yes

3. Has the statistical analysis been performed appropriately and rigorously? 

Reviewer #2: Yes

Reviewer #3: Yes

Reviewer #4: (No Response)

Reviewer #5: Yes

Reviewer #7: Yes

Reviewer #8: Yes

Reviewer #9: Yes

4. Have the authors made all data underlying the findings in their manuscript fully available?

Reviewer #2: Yes

Reviewer #3: No

Reviewer #4: (No Response)

Reviewer #5: No

Reviewer #7: Yes

Reviewer #8: No

Reviewer #9: Yes

5. Is the manuscript presented in an intelligible fashion and written in standard English?

Reviewer #2: Yes

Reviewer #3: Yes

Reviewer #4: (No Response)

Reviewer #5: Yes

Reviewer #7: Yes

Reviewer #8: Yes

Reviewer #9: Yes

6. Review Comments to the Author

Reviewer #2: (No Response)

Reviewer #3: Authors have addressed all the comments though few are clarified why they have mentioned and can't be modified further. Just one comment to author: Reflect the overall findings in conclusion of the abstract, so that your takeaway key message is heard by readers of policy makers of Bhutan.

Reviewer #4: (No Response)

Reviewer #5: Definitely this study will provide a baseline for many more studies that can be carried out in the future.

Reviewer #7: (No Response)

Reviewer #9: All comments of the reviewers have been addressed. Please also read this and cite in the discussion: https://www.taylorfrancis.com/chapters/oa-edit/10.4324/9781003187462-17/negotiating-covid-19-bhutan-mary-grace-pelayo-ian-christopher-rocha-jigme-yoezer

7. PLOS authors have the option to publish the peer review history of their article (what does this mean?). If published, this will include your full peer review and any attached files.

Reviewer #2: **Yes: **Kinley Gyem

Reviewer #3: No

Reviewer #4: No

Reviewer #5: No

Reviewer #7: No

Reviewer #8: **Yes: **Mary Grace Pelayo Arellano

Reviewer #9: No

---

## [Author Response · Author response to Decision Letter 2]

9 Mar 2025

Response to Reviewers’ Comments (email Attachment)

Title: Patient satisfaction in regional referral hospitals of Bhutan: Insights from a cross-sectional study

Well-structured, relevant, and methodologically sound.

Strong use of statistical analysis to determine patient satisfaction predictors.

Areas for Improvement

A. Introduction

Comment: Line 61-62

“However, there is still no consensus on a universally accepted definition of patient satisfaction.”

Comment: What definition was opted for patient satisfaction assessment in Bhutan’s Context for this study?

Response: We sincerely thank the reviewer for their insightful comment. As stated in the manuscript, there is no universally accepted definition of patient satisfaction, as it is multifactorial and subjective. Therefore, this study did not adopt any specific definition. Instead, patient satisfaction was assessed using the Patient Satisfaction Questionnaire Short Form (PSQ-18), a validated and widely used survey tool that measures key domains such as general satisfaction, technical quality, interpersonal manners, communication, financial aspects, time spent with the doctor, and accessibility/convenience. By using this standardized instrument, the study ensured an objective and structured evaluation without the need for an independent definition, facilitating a more practical and comparable assessment in line with established methodologies in patient satisfaction research.

Line 74-92

Comment: The discussion on Bhutan’s healthcare system is informative, but a comparison with other low-resource settings would strengthen the rationale. The introduction does not clearly explain why this study is particularly needed in Bhutan.

Response: We sincerely thank the reviewer for their insightful comment. We chose not to include comparisons with other low-resource settings in the introduction, as we believe such comparisons are more appropriate for the discussion section. Instead, we focused on emphasizing the broader significance of measuring patient satisfaction to establish a strong rationale for this study.

While patient satisfaction evaluation is essential in all healthcare systems globally, the Bhutanese healthcare system is no exception. Assessing customer satisfaction is a fundamental component of continuous quality improvement in any service-oriented sector. Although there is no uniquely urgent need for patient satisfaction studies in Bhutan, we have mentioned in manuscript that the country is in the early stages of developing a quality-oriented healthcare culture. Bhutan has adopted the Bhutan Healthcare Standard for Quality Assurance (BHSQA) as its national healthcare standard, with patient satisfaction as one of its key performance indicators. While other indicators are also important, evaluating patient satisfaction at this foundational stage provides valuable insights into service delivery and areas for improvement, supporting the country's broader efforts to enhance healthcare quality. We hope this clarifies our approach.

B. Methods

Line 111-114

1) “During this period, voluntary feedback was sought from every fifth inpatient and outpatient, aged 18 years and older.”

Comment: Why was every fifth patient chosen? A justification for this method (e.g., randomization, reducing response bias) would strengthen the methodology.

Response: We sincerely thank the reviewer for their insightful comment. As this is a retrospective study, the researchers did not have control over the sampling approach. The survey was part of the hospital’s routine quality assessment, and the sampling method was predetermined by the respective hospitals. The hospitals selected every fifth patient to meet their predefined sample size requirements, which were primarily influenced by patient load, the need for systematic selection, and efforts to reduce selection bias.

We have revised the methodology section accordingly and added the following statement to reflect the hospital’s actual intention:

“A systematic selection of every fifth patient was employed to minimize selection bias and meet the hospital's predefined sample size requirements for internal evaluation”

2) “For illiterate participants, responses were recorded by their friends or quality assurance officials.” This introduces possible response bias since friends or officials may influence responses.

Comment: Discuss whether any measures were taken to mitigate this bias (e.g., using neutral interviewers).

Response: We sincerely thank the reviewer for their valuable comment. We acknowledge that the actual survey conducted by the hospitals did not include specific measures to mitigate response bias. As mentioned earlier in the manuscript, since the survey was administered by the hospitals, it could have introduced a response bias, particularly skewed toward positive responses. To address this, we suggested the implementation of online survey systems or the engagement of external evaluators.

In response to the additional concern raised by the reviewer regarding the potential influence of friends or quality assurance officials when recording responses for illiterate participants, we have amended the discussion section as follows:

“However, it is important to acknowledge that the surveys were administered by the quality units of the respective healthcare centres, and for illiterate participants, responses were recorded with the assistance of friends or quality assurance officials. This process may have introduced response bias. To mitigate this, healthcare centres could consider implementing online, anonymous survey systems or engaging external evaluators to assess patient satisfaction.

We hope this amendment addresses the concern raised.

C. Discussion

Line 239 - 251

"Similarly, educated populations, particularly those with exposure to superior services and familiarity with stringent quality standards, may harbor elevated expectations, potentially resulting in lower satisfaction levels."

Comment: Other factors (e.g., hospital workload, type of medical condition) might also affect satisfaction but are not discussed.

Suggestion: Acknowledge potential confounders and their limitations in interpreting results.

Response: We sincerely thank the reviewer for their insightful comment. Patient satisfaction is a multifactorial concept, and we have emphasized this throughout the manuscript. In our discussion, we focused on the significant predictors identified through multivariate logistic regression, which accounts for confounding factors. Therefore, we did not specifically discuss confounders in relation to education.

Additionally, we highlighted that factors such as patient type, age, sex, and marital status were not significant predictors after adjusting for other variables, underscoring the importance of considering confounding factors.

We hope this response clarifies our reasoning for not explicitly discussing confounding factors in relation to education

D. Conclusion

Line 408 - 409

“As Bhutan's healthcare system evolves, leveraging patient satisfaction data for CQI is crucial.”

This is a general statement that could be more impactful.

Comment: End with a call to action, such as recommending regular nationwide patient satisfaction surveys.

Response: We sincerely appreciate the reviewer’s insightful comment. We have revised the statement as follows:

“As Bhutan's healthcare system evolves, patient satisfaction data should be systematically integrated into CQI efforts. Regular nationwide assessments must be implemented to identify areas for improvement while ensuring a balanced approach to safeguard healthcare professionals' morale and care standards”

Additional Recommendations

Ensure all tables and figures are clearly referenced in the text.

Response: Thank you for the comment, we have again checked ensured that tables and figures are clearly referenced in the text.

Maintain consistency in citation formatting (some references use "et al." while others list all authors).

Response: Thank you for your comment. We have thoroughly reviewed the references and ensured consistency in formatting. For references with more than six authors, we have listed the first six authors followed by "et al." as per the guidelines.

Review Comments to the Author (mentioned in the email)

Reviewer #2: (No Response)

Response: Not required

Reviewer #3: Authors have addressed all the comments though few are clarified why they have mentioned and can't be modified further. Just one comment to author: Reflect the overall findings in conclusion of the abstract, so that your takeaway key message is heard by readers of policy makers of Bhutan.

Response: Thank you for your valuable feedback. We appreciate the suggestion to ensure our key findings are clearly conveyed to policymakers. In response, we have revised the conclusion in the abstract to better reflect the overall findings. The amended version now reads:

“Overall, with a score of 4.06 on a 5-point Likert scale, patient satisfaction in Bhutan is high. However, our findings highlight the need to address socio-demographic disparities in patient satisfaction. As the Bhutanese socio-demographic landscape evolves, satisfaction levels may decline. To enhance overall satisfaction, healthcare policymakers should focus on improving accessibility and convenience. Strategies such as establishing dynamic limits on free services, exploring private sector engagement in advanced healthcare service, and strengthening the healthcare workforce are essential for sustainable and quality healthcare service delivery.”

We hope this revision effectively conveys the key message to policymakers and stakeholders.

Reviewer #4: (No Response)

Response: Not required

Reviewer #5: Definitely this study will provide a baseline for many more studies that can be carried out in the future.

Response: Not required

Reviewer #7: (No Response)

Response: Not required

Reviewer #9: All comments of the reviewers have been addressed. Please also read this and cite in the discussion: https://www.taylorfrancis.com/chapters/oa-edit/10.4324/9781003187462-17/negotiating-covid-19-bhutan-mary-grace-pelayo-ian-christopher-rocha-jigme-yoezer

Response: Thank you for the suggested reference. We have carefully reviewed the article and appreciate its insights. Upon evaluation, we found that the key information relevant to our study has already been incorporated using the latest available references or original source. Therefore, we feel that adding this citation may not provide additional value to the manuscript. However, we sincerely appreciate the recommendation and hope this clarification addresses the reviewer’s suggestion.

---

## [Decision Letter · Decision Letter 2]

Patient satisfaction in regional referral hospitals of Bhutan: Insights from a cross-sectional study.

PONE-D-24-44865R2

Dear Dr. Dorji,

We’re pleased to inform you that your manuscript has been judged scientifically suitable for publication and will be formally accepted for publication once it meets all outstanding technical requirements.

Kind regards,

Ian Christopher N Rocha, MD, MBA, MHSS

Academic Editor

PLOS ONE

Additional Editor Comments (optional):

Reviewers' comments:

Reviewer's Responses to Questions

**Comments to the Author**

1. If the authors have adequately addressed your comments raised in a previous round of review and you feel that this manuscript is now acceptable for publication, you may indicate that here to bypass the “Comments to the Author” section, enter your conflict of interest statement in the “Confidential to Editor” section, and submit your "Accept" recommendation.

Reviewer #2: All comments have been addressed

2. Is the manuscript technically sound, and do the data support the conclusions?

Reviewer #2: Yes

3. Has the statistical analysis been performed appropriately and rigorously? 

Reviewer #2: Yes

4. Have the authors made all data underlying the findings in their manuscript fully available?

Reviewer #2: Yes

5. Is the manuscript presented in an intelligible fashion and written in standard English?

Reviewer #2: Yes

6. Review Comments to the Author

Reviewer #2: Thank you for inviting me to review this manuscript again. I have reviewed the revisions and the authors' responses to my previous comments. At this stage, I have no further substantive feedback to add, as my earlier concerns have been adequately addressed.

If there are specific aspects the journal would like me to re-evaluate, please let me know. Otherwise, I defer to the editorial judgment on whether the manuscript is now suitable for publication.

7. PLOS authors have the option to publish the peer review history of their article (what does this mean?). If published, this will include your full peer review and any attached files.

Reviewer #2: **Yes: **Kinley Gyem

---

## [Editor Report · Acceptance letter]

PONE-D-24-44865R2

PLOS ONE

Dear Dr. Dorji,

I'm pleased to inform you that your manuscript has been deemed suitable for publication in PLOS ONE. Congratulations! Your manuscript is now being handed over to our production team.

Kind regards,

on behalf of

Dr. Ian Christopher N Rocha

Academic Editor

PLOS ONE